# Finite-Time Stability for Caputo Nabla Fractional-Order Switched Linear Systems

**Peng Xu** [1,2], **Fei Long** [2,*], **Qixiang Wang** [1,2], **Ji Tian** [1], **Xiaowu Yang** [2] **and Lipo Mo** [3]

1   School of Electrical Engineering, Guizhou University, Guiyang 550025, China
2   School of Artificial Intelligence and Electrical Engineering, Guizhou Institute of Technology, Special Key Laboratory of Artificial Intelligence and Intelligent Control of Guizhou Province, Guiyang 550003, China
3   School of Mathematics and Statistics, Beijing Technology and Business University, Haidian, Beijing 100048, China
*   Correspondence: feilong@git.edu.cn

**Abstract:** In this paper, we address the finite-time stability problem of Caputo nabla fractional-order switched linear systems with $\alpha \in (0, 1)$. Firstly, the monotonicity of the discrete Mittag-Leffler function is proposed. Secondly, under the per-designed switching rules, the form of the solution for Caputo nabla fractional-order switched linear systems is obtained by using the discrete unit step function. On the above basis, some sufficient conditions of finite-time stability for Caputo nabla fractional-order switched linear systems are proposed, according to the discrete Grönwall inequality and the monotonicity of the discrete Mittag-Leffler function. Finally, simulation verification is carried out via three numerical examples.

**Keywords:** finite-time stability; Caputo nabla fractional-order switched linear systems; discrete unit step function; discrete Mittag-Leffler function

## 1. Introduction

Switched systems are a class of hybrid dynamical systems, which consist of finite subsystems and corresponding switching signals; switching signals coordinate the switching between the subsystems. In the past few decades, the switched systems have drawn a lot of attention and interest in the field of system control, such as network control [1], robot control [2], vehicle control [3], and so on.

Fractional-order dynamical systems can be described by non-integer order differential equations or non-integer difference equations [4]. Numerous analyses, real-world problems, and numerical methods have been solved by fractional derivatives, integrals and differences over the last few decades. For example, fractional-order modeling of the Gemini Virus in Capsicum by fractional calculus obtained optimal control methods for biological control [5]; extensions and additions to image encryption via fractional calculus [6]; and numerous analyses of the controllability results of the non-dense Hilfer neutral fractional derivative [7]. Fractional calculi were developed by Grünwald–Letnikov, Riemann–Liouville, and Caputo in the past epoch. Fractional-order differential equations were proposed in 1695, while fractional-order difference equations were introduced in 1974 [8]. Furthermore, fractional-order discrete-time dynamics systems have also emerged with affluent results. For example, the stability of fractional-order difference systems have been discussed in [9–11], and many researchers have handled a lot of contents about different types of fractional-order difference operators [9,12].

With the maturity of fractional calculus theory, the research on fractional-order systems has become more and more perfect. For example, Wei et al. proposed the Mittag-Leffler stability of fractional difference dynamic systems [13], and Wu et al. proposed the finite-time stabilization of fractional-order discrete time-delay systems [14]. Fractional calculus theory has been widely used in heat conduction [15], capacitance [16] and other fields,

especially in the field of systems control, and outstanding results based on fractional calculus theory have appeared.

Recently, some researchers have combined fractional-order theory with switched systems, and established fractional-order switched system models to solve multiple problems that cannot be solved by using integer-order switched systems, such as viscoelastic systems [17], circuit models [18], quantum mechanics [19], economics system [20], electrode-electrolyte polarization [21], etc. At present, the majority of the research on fractional-order switched systems is focused mainly on stability, such as finite-time stability [22], asymptotic stability [23], exponential stability [24], external stability [25], and so on. However, it is noteworthy that most of research results stay on fractional-order continuous-time switched systems, and there are few results on the stability of fractional-order discrete-time switched systems.

It is well-known that the stability of integer-order discrete-time switched systems is frequently discussed by using the common Lyapunov method [26], the multi-Lyapunov method [27], and the dwell time method [28]. In comparison, the multi-Lyapunov method is generally less conservative than the common Lyapunov method. The dwell time method is further divided into the average dwell time method [29], the mode-dependent average dwell time method [28], and the weighted average dwell time method [30]. Lyapunov asymptotic stability describes the steady-state behavior of the system in the infinite-time domain, but in some actual conditions, many systems run in a finite-time domain or the transient behaviors of the systems in a finite-time domain need to be considered in practice [31]. Therefore, in this paper, we study the finite-time stability for fractional-order discrete-time switched systems.

Generally speaking, the lower bounds of the Caputo fractional difference operator and the fractional sum operator must be consistent when taking the sum of fractional-orders on both sides of the Caputo fractional difference equation. However, the lower bound of the fractional-order switched system cannot be updated with the occurrence of switching due to the memory [32], hereditary [33], and non-locality [34] of the fractional-order system. As a result, the state trajectory of Caputo fractional-order discrete-time switched systems cannot follow the expression of integer-order discrete-time switched systems, which is also the most essential difference between fractional-order switched systems and integer-order switched systems. Because a fractional-order switched system has a different system matrix after each switching, its solution also cannot be obtained directly through the system equation and Mittag-Leffler function as in [35], and the study of its finite-time stability cannot be obtained simply through the Riemann–Liouville nabla properties and the generalized Grönwall inequality as in [36]. To our best knowledge, there are few systematic results on the stability of fractional discrete-time switched systems due to the above essential differences. Therefore, it is very meaningful and challenging to study the stability of fractional-order discrete-time switched systems.

This paper discusses the finite-time stability problem of Caputo nabla fractional-order switched linear systems. The main contributions of this paper are summarized as follows:

i.  In order to overcome the above-mentioned problems, we provide the monotonicity of the discrete Mittag-Leffler function, and obtained the expression of a solution for a Caputo nabla fractional-order switched linear system by using the discrete unit step function.

ii. The sufficient conditions of finite-time stability for a Caputo nabla fractional-order switched linear system are provided based on the discrete Grönwall inequality and the monotonicity of the discrete Mittag-Leffler function.

The structure of this paper is as follows: in Section 2, we recalled some important definitions and lemmas; we re-describe the Caputo nabla fractional-order switched linear system; we propose the finite-time solution and finite-time stability conditions of the Caputo nabla fractional-order switched linear system in Section 3; in Section 4, the feasibility of finite-time stability conditions is verified by using three numerical examples; and finally, the article is summarized in Section 5.

**Notations**: $\mathbb{N}_a = \{a, a+1, a+2, \ldots\}, \mathbb{N}^b = \{\ldots, b-2, b-1, b\}, \mathbb{N}_a^b = \{a, a+1, a+2, \ldots, b-1, b\}$ represents the time set, $\binom{p}{q} = \frac{\Gamma(p+1)}{[\Gamma(q+1)\Gamma(p-q+1)]}$ is the generalized binomial coefficient, $p^{\overline{q}} = \frac{\Gamma(p+q)}{\Gamma(p)}$ is the rising function. For a given vector $x \in \mathbb{R}^n$, $\|x\|$ stands for the Euclidean norm, where $\|x\| = \sqrt{\sum_{i=1}^n x_i^2}$. For a given matrix $A \in \mathbb{R}^{n \times s}$, $A^T$ represents the transpose of the matrix A, $\|A\|$ represents the spectral norm, which is defined by $\|A\| = \sqrt{\lambda_{\max}(A^T A)}$. The discrete unit step function is described by $\overline{H}(k) = \begin{cases} 0, k \in \mathbb{N}^{-1} \\ 1, k \in \mathbb{N}_0 \end{cases}$.

## 2. Preliminaries

Some important definitions, lemmas, and the discrete Mittag-Leffler function are introduced in this section.

**Definition 1** ([8]). *For the function $f(\cdot) : \mathbb{N}_{b+1-n} \to \mathbb{R}$, its nth integer order backward difference is defined by:*

$$\nabla^n f(k) = \sum_{i=0}^n (-1)^i \binom{n}{i} f(k-i).$$

*Its αth Grünwald–Letnikov difference is defined by:*

$$_b^G \nabla_k^\alpha f(k) = \sum_{i=0}^{k-b-1} (-1)^i \binom{\alpha}{i} f(k-i).$$

*Its αth Caputo fractional difference is defined by:*

$$_b^C \nabla_k^\alpha f(k) = {}_b^G \nabla_k^{\alpha-n}(\nabla^n f(k)),$$

*where $\alpha \in (n-1, n)$, $n \in \mathbb{N}_0$, $k \in \mathbb{N}_{b+1}$ and $b \in \mathbb{R}$.*

**Remark 1.** *In general, there are three widely-used definitions of the fractional difference operators, i.e., the Grünwald–Letnikov operator, the Riemann–Liouville operator, and the Caputo operator. The Caputo operator can be seen as an improvement on the Grünwald–Letnikov fractional-order operator; this type definition is convenient for solving the problem of the initial edge value of fractional-order difference equations, and this expression can also be interpreted in the frequency domain through the Laplace or Fourier transform, to describe the practicality of the interpretations for initial conditions. It thus lays solid foundations for the effective applications of fractional calculus in the field of engineering. In this paper, considering the numerous Caputo operator's advantages, the Caputo nabla operator is adopted.*

**Definition 2** ([37]). *The αth fractional sum of the function $f(\cdot) : \mathbb{N}_{b+1-n} \to \mathbb{R}$ is defined by:*

$$_b \nabla_k^{-\alpha} f(k) = \frac{1}{\Gamma(\alpha)} \sum_{s=b+1}^k (k-s+1)^{\overline{\alpha-1}} f(s),$$

*where $\alpha \in (n-1, n)$, $k \in \mathbb{N}_{b+1}$, $s \in \mathbb{N}_{b+1}^k$ and $b \in \mathbb{R}$. $\Gamma(\alpha) := \int_0^{+\infty} t^{\alpha-1} e^{-t} dt$ represents the gamma function.*

**Lemma 1** ([38]). *Let $0 < \alpha < 1$ and $f(\cdot) : \mathbb{N}_b \to \mathbb{R}^n$. Then for arbitrary $b \in \mathbb{R}$:*

$$_b \nabla_k^{-\alpha} \left( {}_b^C \nabla_k^\alpha f(k) \right) = f(k) - f(b).$$

**Lemma 2** ([39])**.** *(Discrete Grönwall inequality) Let $f(k)$, $g(k)$ be non-negative, non-decreasing real value functions over the set $J = [b+1, b+T] \cap \mathbb{N}_{b+1}$. Let $h(k)$ be a non-negative function on $J$ and $g(k) < 1$ for $k \in J$. Suppose $0 < \alpha < 1$, the following nabla fractional inequality holds:*

$$h(k) \leq f(k) + g(k)\left(_b\nabla_k^{-\alpha}h(k)\right), k \in J,$$

*Then*

$$h(k) \leq f(k)\sum_{j=0}^{\infty} g^j(k)H_{j\alpha}(k,b), k \in J,$$

*where $H_{j\alpha}(k,b) = (k-b)^{\overline{j\alpha}}/\Gamma(j\alpha+1)$ denotes the fractional discrete Taylor monomial.*

**Remark 2.** *Since some of the conditions of the original version of the generalized Grönwall inequality do not need to be considered in this paper, they are discarded and modified in this paper, resulting in a slightly different citation from the original version, but this does not affect the validity of the inequality, and for a similar citation we can see ([36], lemma 2.6).*

**Definition 3** ([40])**.** *The discrete Mittag-Leffler function is defined as:*

$$F_{\alpha,\beta}(\lambda,k) = \sum_{i=0}^{+\infty} \lambda^i \frac{k^{\overline{i\alpha+\beta-1}}}{\Gamma(i\alpha+\beta)},$$

*where $\alpha > 0$, $\beta \in \mathbb{R}$, $\lambda \in \mathbb{R}$ and $k \in \mathbb{N}_b (b \in \mathbb{R})$.*

**Lemma 3.** *With two fixed parameters $\alpha \in (0,1)$ and $\lambda \in (0,1)$, the discrete Mittag-Leffler function $F_{\alpha,1}(\lambda,k)$ is monotonically increasing with respect to $k \in \mathbb{N}_b(b \in \mathbb{R})$.*

**Proof of Lemma 3.** By view of $\Gamma(\alpha) = \int_0^{+\infty} t^{\alpha-1}e^{-t}dt$ and $p^{\overline{q}} = \Gamma(p+q)/\Gamma(p)$, we have

$$\nabla k^{\overline{\alpha}} = k^{\overline{\alpha}} - (k-1)^{\overline{\alpha}} = \frac{\Gamma(k+\alpha)}{\Gamma(k)} - \frac{\Gamma(k-1+\alpha)}{\Gamma(k-1)} = \frac{\Gamma(k+\alpha)-(k-1)\Gamma(k-1+\alpha)}{\Gamma(k)}$$
$$= \frac{(k+\alpha-1)\Gamma(k+\alpha-1)-(k-1)\Gamma(k-1+\alpha)}{\Gamma(k)} = \frac{\alpha\Gamma(k+\alpha-1)}{\Gamma(k)} = \alpha k^{\overline{\alpha-1}}.$$

According to the Definition 3 and the above equality, we can obtain the follow equality:

$$
\begin{aligned}
\nabla F_{\alpha,1}(\lambda,k) &= F_{\alpha,1}(\lambda,k) - F_{\alpha,1}(\lambda,k-1) = \sum_{i=0}^{+\infty} \lambda^i \frac{k^{\overline{i\alpha}}}{\Gamma(i\alpha+1)} - \sum_{i=0}^{+\infty} \lambda^i \frac{(k-1)^{\overline{i\alpha}}}{\Gamma(i\alpha+1)} \\
&= \sum_{i=0}^{+\infty} \frac{\lambda^i}{\Gamma(i\alpha+1)}\nabla k^{\overline{i\alpha}} = \sum_{i=0}^{+\infty} \frac{\lambda^i}{\Gamma(i\alpha+1)}i\alpha k^{\overline{i\alpha-1}} = \sum_{i=1}^{+\infty} \frac{\lambda^i}{\left(\frac{\Gamma(i\alpha+1)}{i\alpha}\right)}k^{\overline{i\alpha-1}} \\
&= \sum_{i=1}^{+\infty} \frac{\lambda^i}{\Gamma(i\alpha)}k^{\overline{i\alpha-1}} = \sum_{i=0}^{+\infty} \frac{\lambda^{i+1}}{\Gamma((i+1)\alpha)}k^{\overline{(i+1)\alpha-1}} = \lambda\sum_{i=0}^{+\infty} \frac{\lambda^i}{\Gamma(i\alpha+\alpha)}k^{\overline{i\alpha+\alpha-1}} \\
&= \lambda F_{\alpha,\alpha}(\lambda,k).
\end{aligned}
$$

It is obvious that the discrete Mittag-Leffler function $F_{\alpha,\alpha}(\lambda,k)$ is strictly positive with respect to $k \in \mathbb{N}_b(b \in \mathbb{R})$ for the two parameters $\alpha \in (0,1)$ and $\lambda \in (0,1)$ (see [41], Remark 1). As a result, $\nabla F_{\alpha,1}(\lambda,k) = \lambda F_{\alpha,\alpha}(\lambda,k) > 0$ for all $k \in \mathbb{N}_b(b \in \mathbb{R})$. Therefore, the function $F_{\alpha,1}(\lambda,k)$ is monotonically increasing with respect to $k \in \mathbb{N}_b(b \in \mathbb{R})$. This completes the proof. $\square$

Now, a simple number is given to verify the Lemma 3.

**Example 1.** *Consider the discrete Mittag-Leffler function $F_{\alpha,1}(\lambda,k)$ with $\lambda = 0.5$. Figure 1 shows how discrete Mittag-Leffler functions $F_{0.25,1}(0.5,k)$, $F_{0.5,1}(0.5,k)$, $F_{0.75,1}(0.5,k)$, and $F_{0.9,1}(0.5,k)$ have increased over the set $\mathbb{N}_0^{10}$. Furthermore, it is obvious that the larger parameter $\alpha \in (0,1)$, the faster $F_{\alpha,1}(0.5,k)$ increases.*

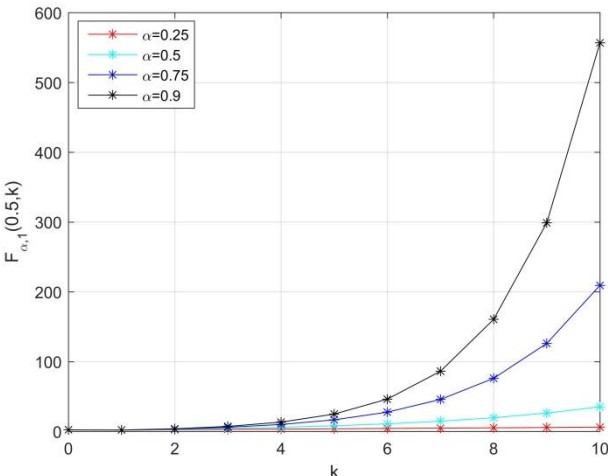

**Figure 1.** Trend of the discrete Mittag-Leffler function $F_{\alpha,1}(0.5, k)$.

## 3. Main Results

Consider the following Caputo nabla fractional-order switched linear system:

$$\begin{cases} {}^{C}_{k_0}\nabla^{\alpha}_k x(k) = A_{\sigma(k)}x(k), \\ x(k_0) = x_0, \end{cases} \tag{1}$$

where $0 < \alpha < 1$; $x(k_0) = x_0$ is the initial state; $x(k) \in \mathbb{R}^n$ denotes the system state. The function $\sigma(k) : \mathbb{N}_{k_0} \to \overline{N} = \{1, 2, \cdots, N\}$ represents the switching signal, which is a right continuous piecewise constant function, and $N$ denotes the total number of systems (to avoid unnecessary situations, we assumed that $N \geq 2$). $A_i \in \mathbb{R}^{n \times n} (i \in \overline{N})$ stands for the known system matrices. Suppose that $\{(i_0, k_0), (i_1, k_1), \cdots, (i_m, k_m), \cdots\}$ is a switching signal $\sigma(k)$ over the interval $\mathbb{N}_{k_0}$, where $\{k_0, k_1, \cdots, k_m, \cdots\}$ denotes a switching time sequence. That is to say, $\sigma(k) = i_m$ as $k \in \mathbb{N}_{k_m}^{k_{m+1}-1}$, in another words, the $k_m$-th subsystem ${}^{C}_{k_0}\nabla^{\alpha}_k x(k) = A_{i_m}x(k)$ is active in the interval $\mathbb{N}_{k_m}^{k_{m+1}-1}$.

Our objective is to find some sufficient conditions that the Caputo nabla fractional-order switched linear system (1) is finite-time stable under pre-constructed switching rules. As a result, the definition of finite-time stability and lemmas are used in discussing finite-time stability conditions.

**Definition 4.** *(Finite-time stable). For three given finite positive scalars $T$, $c_1$ and $c_2(> c_1)$, the Caputo nabla fractional-order switched linear system (1) is said to be finite-time stable with respect to $(c_1, c_2, T)$, if $\|x_0\| \leq c_1$ implies $\|x(k)\| < c_2$ for all $k \in \mathbb{N}_{k_0}^T$.*

Let $\{(\sigma(k_0), k_0), (\sigma(k_1), k_1), \cdots, (\sigma(k_m), k_m)\}$ stand for the switching sequence of $\sigma(k)$ over the set $\mathbb{N}_{k_0}^T$, where $0 \leq k_0 < k_1 < \cdots < k_m \leq T < \infty$. Then the system (1) can be re-described as follows under the above switching sequence:

$$\begin{cases} {}^{C}_{k_0}\nabla^{\alpha}_k x(k) = A_{\sigma(k_l)}x(k), k \in \mathbb{N}_{k_l}^{k_{l+1}-1}(l = 0, 1, \cdots, m-1), \\ {}^{C}_{k_0}\nabla^{\alpha}_k x(k) = A_{\sigma(k_m)}x(k), k \in \mathbb{N}_{k_m}^T, \\ x(k_0) = x_0. \end{cases} \tag{2}$$

Now, the existence of a solution over the set $\mathbb{N}_{k_0}^T$ for the system (2) is provided by using the discrete unit step function $\overline{H}(k)$.

**Lemma 4.** *For $\alpha \in (0,1)$ and $k \in \mathbb{N}_{k_0}^T$, the solution of the Caputo nabla fractional-order switched linear system (2) is given by:*

$$
x(k) = \begin{cases}
x_0 + \frac{1}{\Gamma(\alpha)} \sum\limits_{s=k_0+1}^{k} (k-s+1)^{\overline{\alpha-1}} A_{\sigma(k_0)} x(s), \ k \in \mathbb{N}_{k_0}^{k_1-1}, \\
\vdots \\
x_0 + \frac{1}{\Gamma(\alpha)} \sum\limits_{s=k_0+1}^{k} (k-s+1)^{\overline{\alpha-1}} A_{\sigma(k_0)} x(s) + \sum\limits_{i=1}^{l} \left[ \frac{1}{\Gamma(\alpha)} \sum\limits_{s=k_i+1}^{k} (k-s+1)^{\overline{\alpha-1}} \left( A_{\sigma(k_i)} - A_{\sigma(k_{i-1})} \right) \right] x(s), k \in \mathbb{N}_{k_l}^{k_{l+1}-1}, \\
\vdots \\
x_0 + \frac{1}{\Gamma(\alpha)} \sum\limits_{s=k_0+1}^{k} (k-s+1)^{\overline{\alpha-1}} A_{\sigma(k_0)} x(s) + \sum\limits_{i=1}^{m} \left[ \frac{1}{\Gamma(\alpha)} \sum\limits_{s=k_i+1}^{k} (k-s+1)^{\overline{\alpha-1}} \left( A_{\sigma(k_i)} - A_{\sigma(k_{i-1})} \right) \right] x(s), k \in \mathbb{N}_{k_m}^{T}.
\end{cases}
$$

**Proof of Lemma 4.** For $k \in \mathbb{N}_{k_0}^{k_1-1}$, according to (2), we have:

$$
{}_{k_0}^{C}\nabla_k^{\alpha} x(k) = A_{\sigma(k_0)} x(k). \tag{3}
$$

Taking the fractional sum on both sides of Equation (3) from $k_0$ to $k$, and in view of Lemma 1 and Definition 2, it is obvious that the following equality holds:

$$
x(k) = x_0 + \frac{1}{\Gamma(\alpha)} \sum_{s=k_0+1}^{k} (k-s+1)^{\overline{\alpha-1}} A_{\sigma(k_0)} x(s). \tag{4}
$$

For $k \in \mathbb{N}_{k_1}^{k_2-1}$, by means of Equation (2), the following equality holds by using the discrete unit step function $\overline{H}(k)$:

$$
{}_{k_0}^{C}\nabla_k^{\alpha} x(k) = A_{\sigma(k_0)} x(k) \times \left[ \overline{H}(k) - \overline{H}(k-k_1) \right] + A_{\sigma(k_1)} x(k) \times \overline{H}(k-k_1). \tag{5}
$$

In fact, according to the definition of the discrete unit step function, it is obvious for arbitrary $k \in \mathbb{N}_{k_1}^{k_2-1}$ that $\overline{H}(k-k_1) = 1$ and $\overline{H}(k) - \overline{H}(k-k_1) = 0$. That is to say, Equation (5) is equivalent to Equation (2) as $k \in \mathbb{N}_{k_1}^{k_2-1}$. Furthermore, Equation (5) can be rewritten as Equation (3) when $k \in \mathbb{N}_{k_0}^{k_1-1}$.

Taking the fractional sum on both sides of Equation (5) from $k_0$ to $k$, and in view of Lemma 1 and Definition 2, we can get:

$$
x(k) - x_0 = \frac{1}{\Gamma(\alpha)} \sum_{s=k_0+1}^{k} (k-s+1)^{\overline{\alpha-1}} A_{\sigma(k_0)} x(s) - \frac{1}{\Gamma(\alpha)} \sum_{s=k_1+1}^{k} (k-s+1)^{\overline{\alpha-1}} A_{\sigma(k_0)} x(s) +
$$
$$
\frac{1}{\Gamma(\alpha)} \sum_{s=k_1+1}^{k} (k-s+1)^{\overline{\alpha-1}} A_{\sigma(k_1)} x(s).
$$

That is:

$$
x(k) = x_0 + \frac{1}{\Gamma(\alpha)} \sum_{s=k_0+1}^{k} (k-s+1)^{\overline{\alpha-1}} A_{\sigma(k_0)} x(s) + \frac{1}{\Gamma(\alpha)} \sum_{s=k_1+1}^{k} (k-s+1)^{\overline{\alpha-1}} (A_{\sigma(k_1)} - A_{\sigma(k_0)}) x(s). \tag{6}
$$

Using a similar technique to that mentioned above, extending to $k \in \mathbb{N}_{k_l}^{k_{l+1}-1}$, gives us:

$$
x(k) = x_0 + \frac{1}{\Gamma(\alpha)} \sum_{s=k_0+1}^{k} (k-s+1)^{\overline{\alpha-1}} A_{\sigma(k_0)} x(s) + \sum_{i=1}^{l} \left[ \frac{1}{\Gamma(\alpha)} \sum_{s=k_i+1}^{k} (k-s+1)^{\overline{\alpha-1}} \left( A_{\sigma(k_i)} - A_{\sigma(k_{i-1})} \right) \right] x(s).
$$

and by extending to $k \in \mathbb{N}_{k_m}^T$, it is obvious that:

$$x(k) = x_0 + \frac{1}{\Gamma(\alpha)} \sum_{s=k_0+1}^{k} (k-s+1)^{\overline{\alpha-1}} A_{\sigma(k_0)} x(s) + \sum_{i=1}^{m} \left[ \frac{1}{\Gamma(\alpha)} \sum_{s=k_i+1}^{k} (k-s+1)^{\overline{\alpha-1}} \left( A_{\sigma(k_i)} - A_{\sigma(k_{i-1})} \right) \right] x(s).$$

Therefore, the conclusion of this lemma holds. This completes the proof. □

**Remark 3.** *The steps described above refer to firstly using the discrete unit step function to re-describe each interval of the fractional difference equation to ensure that it satisfies the form represented by (5), and then the fractional-order sum $_{k_0}\nabla_k^{-\alpha}$ on both sides of equations by using Lemma 1 and Definition 2. The initial moment of its fractional-order sum is always taken as $k_0$, which even accounts for the initial moment of the current interval when it does not start from $k_0$, the state of the current system is still affected by the specific parameters in the previous interval.*

Now, we use Lemma 4 to give the finite-time stability condition of a Caputo nabla fractional-order switched linear system (1).

**Theorem 1.** *The Caputo nabla fractional-order switched linear system (1) is finite-time stable with respect to the triplet $(c_1, c_2, T)$. For two positive finite real numbers $c_1, T$, there exist a positive finite real number $c_2 (> c_1)$ such that the following conditions hold:*

$$2(m+1)r < 1, \tag{7}$$

$$c_2 > c_1 F_{\alpha,1}(2(m+1)r, T-k_0), \tag{8}$$

*where $r = \max\{\|A_1\|, \|A_2\|, \cdots, \|A_N\|\}$, and $m$ denotes the switching number of $\sigma(k)$ over the set $\mathbb{N}_{k_0}^T$.*

**Proof of Theorem 1.** Suppose that $\{(\sigma(k_0), k_0), (\sigma(k_1), k_1), \cdots, (\sigma(k_m), k_m)\}$ is a switching sequence of $\sigma(k)$ over the set $\mathbb{N}_{k_0}^T$, that is to say, $\mathbb{N}_{k_0}^T = \mathbb{N}_{k_0}^{k_1-1} \cup \mathbb{N}_{k_1}^{k_2-1} \cup \cdots \cup \mathbb{N}_{k_{m-1}}^{k_m-1} \cup \mathbb{N}_{k_m}^T$ and:

$$\sigma(k) = \begin{cases} \sigma(k_i), & k \in \mathbb{N}_{k_i}^{k_i-1}, \ i = 0, 1, \cdots, m-1, \\ \sigma(k_m), & k \in \mathbb{N}_{k_m}^T. \end{cases} \tag{9}$$

Then system (1) can be described as system (2) under switching rule (9).

For $k \in \mathbb{N}_{k_0}^{k_1-1}$, in view of Lemma 4, the following equality is obvious:

$$x(k) = x_0 + \frac{1}{\Gamma(\alpha)} \sum_{s=k_0+1}^{k} (k-s+1)^{\overline{\alpha-1}} A_{\sigma(k_0)} x(s). \tag{10}$$

Taking the norm on both sides of (10), the following inequality is obvious according to the definition of the fractional sum (Definition 2):

$$\begin{aligned} \|x(k)\| &\leq \|x_0\| + \frac{\|A_{\sigma(k_0)}\|}{\Gamma(\alpha)} \sum_{s=k_0+1}^{k} (k-s+1)^{\overline{\alpha-1}} \|x(s)\| \\ &\leq \|x_0\| + \frac{r}{\Gamma(\alpha)} \sum_{s=k_0+1}^{k} (k-s+1)^{\overline{\alpha-1}} \|x(s)\| \\ &\leq \|x_0\| + 2r \,_{k_0}\nabla_k^{-\alpha} \|x(k)\|, \end{aligned} \tag{11}$$

where $r = \max\{\|A_1\|, \|A_2\|, \cdots, \|A_N\|\}$.

According to condition (7), $2r < 1$. Furthermore, based on the discrete Grönwall inequality (Lemma 2) and $2r < 1$, it can be obtained easily that (11) implies the following inequality:

$$\|x(k)\| \le \|x_0\| \sum_{j=0}^{\infty} (2r)^j H_{j\alpha}(k, k_0) = \|x_0\| F_{\alpha,1}(2r, k - k_0) = \|x_0\| F_{\alpha,1}(2(0+1)r, k - k_0) \tag{12}$$

For $k \in \mathbb{N}_{k_l}^{k_{l+1}-1}(l = 1, 2, \cdots, m-1)$, by means of Lemma 4, we have:

$$x(k) = x_0 + \frac{1}{\Gamma(\alpha)} \sum_{s=k_0+1}^{k} (k-s+1)^{\overline{\alpha-1}} A_{\sigma(k_0)} x(s) + \sum_{i=1}^{l} \left[ \frac{1}{\Gamma(\alpha)} \sum_{s=k_i+1}^{k} (k-s+1)^{\overline{\alpha-1}} \left( A_{\sigma(k_i)} - A_{\sigma(k_{i-1})} \right) \right] x(s). \tag{13}$$

Taking the norm on both sides of (13), the following inequality can be obtained according to the definition of the fractional sum (Definition 2):

$$\begin{aligned}
\|x(k)\| &\le \|x_0\| + \frac{\|A_{\sigma(k_0)}\|}{\Gamma(\alpha)} \sum_{s=k_0+1}^{k} (k-s+1)^{\overline{\alpha-1}} \|x(s)\| + \sum_{i=1}^{l} \left[ \frac{\|A_{\sigma(k_i)}\| + \|A_{\sigma(k_{i-1})}\|}{\Gamma(\alpha)} \sum_{s=k_i+1}^{k} (k-s+1)^{\overline{\alpha-1}} \right] \|x(s)\| \\
&\le \|x_0\| + r \frac{1}{\Gamma(\alpha)} \sum_{s=k_0+1}^{k} (k-s+1)^{\overline{\alpha-1}} \|x(s)\| + \sum_{i=1}^{l} \left[ 2r \frac{1}{\Gamma(\alpha)} \sum_{s=k_i+1}^{k} (k-s+1)^{\overline{\alpha-1}} \right] \|x(s)\| \\
&\le \|x_0\| + 2(l+1)r \frac{1}{\Gamma(\alpha)} \sum_{s=k_0+1}^{k} (k-s+1)^{\overline{\alpha-1}} \|x(s)\| \\
&\le \|x_0\| + 2(l+1)r \left( {}_{k_0}\nabla_k^{-\alpha} \|x(k)\| \right),
\end{aligned} \tag{14}$$

where $r = \max\{\|A_1\|, \|A_2\|, \cdots, \|A_N\|\}$.

According to condition (7), $2(l+1)r < 1$. Furthermore, based on the discrete Grönwall inequality (Lemma 2) and $2(l+1)r < 1$, it can be obtained easily that (14) implies the following inequality:

$$\|x(k)\| \le \|x_0\| \sum_{j=0}^{\infty} [2(l+1)r]^j H_{j\alpha}(k, k_0) = \|x_0\| F_{\alpha,1}(2(l+1)r, k - k_0). \tag{15}$$

Following the same steps, and based on Lemma 3, and extending to $k \in \mathbb{N}_{k_m}^T$, it can be obtained that:

$$\|x(k)\| \le \|x_0\| F_{\alpha,1}(2(m+1)r, k - k_0) \le \|x_0\| F_{\alpha,1}(2(m+1)r, T - k_0). \tag{16}$$

Based on condition (8), and the inequalities (12), (15) and (16), it is obvious that $\|x_0\| \le c_1$ implies $\|x(k)\| < c_2$ for all $k \in \mathbb{N}_{k_0}^T$. That is to say, the Caputo nabla fractional-order switched linear system (1) is finite-time stable with respect to the triplet $(c_1, c_2, T)$. This completes the proof. $\square$

## 4. Numerical Examples

**Example 2.** *Consider the following Caputo nabla fractional-order switched linear system.*

$$\begin{aligned}
{}_{k_0}^C\nabla_k^{0.5} x(k) = \begin{cases} 0.08x(k), 0 \le k < 3, \\ 0.06x(k), 3 \le k < 5, \\ 0.04x(k), 5 \le k \le 8, \end{cases}
\end{aligned} \tag{17}$$

*where $k_0 = 0$, $x(k_0) = 0.125$.*

Select $c_1 = 0.125$, $c_2 = 2$, and $T = 8$, and it is obvious for system (17) that $r = 0.08$ and $m = 2$. It can be checked that:

$$2(m+1)r = 0.48 < 1, c_2 > c_1 F_{\alpha,1}(2(m+1)r, T - k_0) = 0.125 \times F_{0.5,1}(0.48, 8) = 1.987025 < 2.$$

That is, the conditions (7) and (8) of Theorem 1 hold. The corresponding simulation is carried out for system (17) with $x(0) = 0.125$, and it is easy to obtain from Figure 2 that the value of $\|x(k)\|$ does not exceed the given threshold $c_2 = 2$ over 0 to 8 s, which validates that under the sufficient conditions in Theorem 1, system (17) is finite-time stable with respect to $(0.125, 2, 8)$.

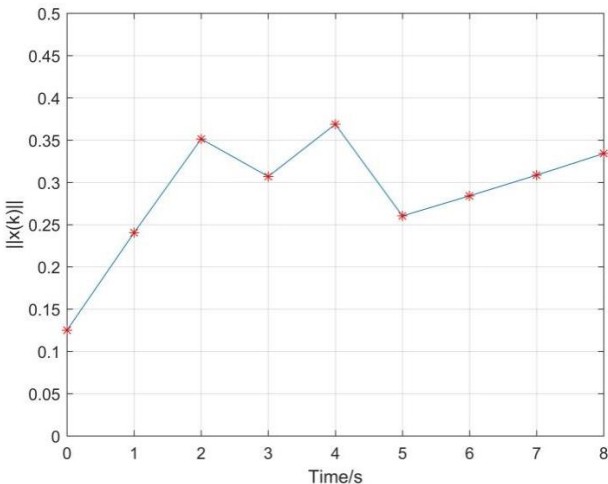

**Figure 2.** The trajectory of $\|x(k)\|$.

**Example 3.** *Consider the following Caputo nabla fractional-order switched linear system.*

$$\begin{cases} {}^{C}_{k_0}\nabla^{0.4}_k x(k) = A_{\sigma(k)}x(k), \\ x(k_0) = x_0, \end{cases} \tag{18}$$

*where* $k_0 = 0$, $x(k_0) = \begin{bmatrix} 0.06 & 0.05 \end{bmatrix}^T$,

$$A_1 = \begin{bmatrix} 0.06 & -0.02 \\ 0.02 & 0.05 \end{bmatrix}, \; A_2 = \begin{bmatrix} -0.01 & 0.05 \\ -0.02 & 0.04 \end{bmatrix}, \; A_3 = \begin{bmatrix} 0.04 & 0.02 \\ 0.05 & 0.06 \end{bmatrix}.$$

$c_1 = 0.08$, $c_2 = 3.745$ and $T = 8$, and by some straightforward calculations, it is obvious for system (18) that $r = 0.078$. Suppose that the switching number of $\sigma(k)$ over the set $\mathbb{N}_0^8$ is 3, i.e., $m = 3$. It can be checked that:

$$2(m+1)r = 0.624 < 1, c_2 > c_1 F_{\alpha,1}(2(m+1)r, T - k_0) = 0.08 \times F_{0.4,1}(0.624, 8) = 3.735304 < 3.745.$$

Figure 3 depicts the switching signal $\sigma(k)$, and it is easy to find that the different subsystems possess different dwell times. The time response of the considered system state $x(k)$ under the switching signal $\sigma(k)$ is shown in Figure 4 (left). Furthermore, Figure 4 (right) illustrates the trajectory of $\|x(k)\|$, which satisfies the given initial condition $\|x(0)\| \leq 0.08$. It is easily calculated that $\|x(k)\| < 3.745$ for all $k \in \mathbb{N}_0^8$. Thus, the conducted simulations verify that the considered system is finite-time stable with respect to $(0.08, 3.745, 8)$ if it obeys the given restrictions.

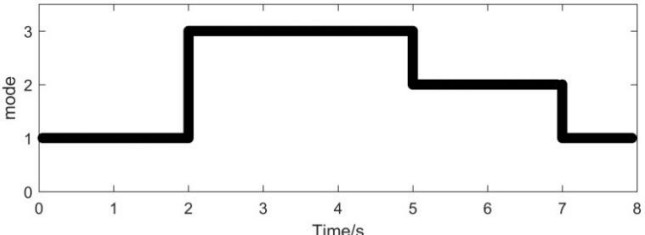

**Figure 3.** The switching signals $\sigma(k)$.

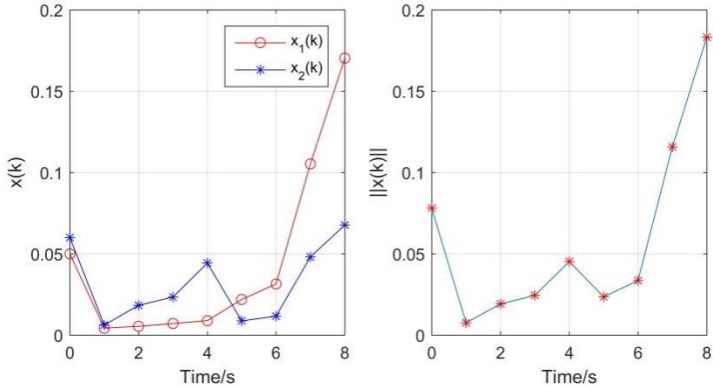

**Figure 4.** State trajectories of $x(k)$ (**left**) and trajectories of $\|x(k)\|$ (**right**).

Therefore, if conditions (7) and (8) of Theorem 1 are satisfied, system (18) is finite-time stable with respect to the triplet $(0.08, 3.745, 8)$.

**Example 4.** Consider the following Caputo nabla fractional-order switched linear system.

$$\begin{cases} {}_{k_0}^{C}\nabla_k^{0.35}x(k) = A_{\sigma(k)}x(k), \\ \qquad\qquad x(k_0) = x_0, \end{cases} \tag{19}$$

where $k_0 = 0$, $x(k_0) = \begin{bmatrix} 0.08 & 0.05 \end{bmatrix}^T$.

$$A_1 = \begin{bmatrix} 0.06 & -0.02 \\ 0.02 & 0.05 \end{bmatrix}, \ A_2 = \begin{bmatrix} 0.05 & 0.06 \\ -0.06 & 0.03 \end{bmatrix}, \ A_3 = \begin{bmatrix} 0.05 & 0.02 \\ -0.01 & 0.04 \end{bmatrix}.$$

Select $c_1 = 0.095$, $c_2 = 3.5$, $T = 8$, and it is obvious for system (19) that $r = 0.045$. Suppose the switching number of $\sigma(k)$ over the set $\mathbb{N}_0^8$ is five, i.e., $m = 5$ (see Figure 5). After a straightforward calculation, it can be checked that:

$$2(m+1)r = 0.54 < 1, c_2 > c_1 F_{\alpha,1}(2(m+1)r, T - k_0) = 0.095 \times F_{0.35,1}(0.54, 8) = 1.15664 < 3.5.$$

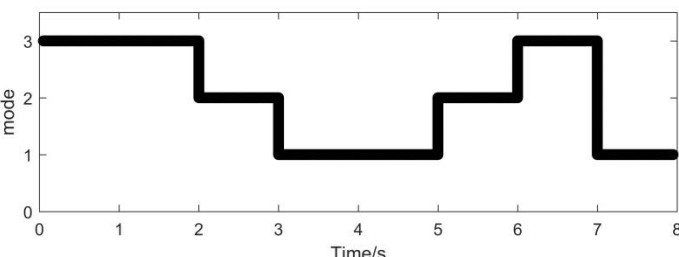

**Figure 5.** The switching signals $\sigma(k)$.

Thus conditions (7) and (8) of Theorem 1 hold. The time response of the considered system state $x(k)$ under the switching signal $\sigma(k)$ is shown in Figure 6 (left) Furthermore,

Figure 6 (right) illustrates the trajectory of $\|x(k)\|$ which satisfies the given initial condition $\|x(0)\| < 0.095$. This shows that $\|x(k)\| < 3.5$, $\forall k \in \mathbb{N}_0^8$. Therefore, the above simulation validates that under sufficient conditions in Theorem 1, system (19) is finite-time stable with respect to the triplet $(0.095, 3.5, 8)$.

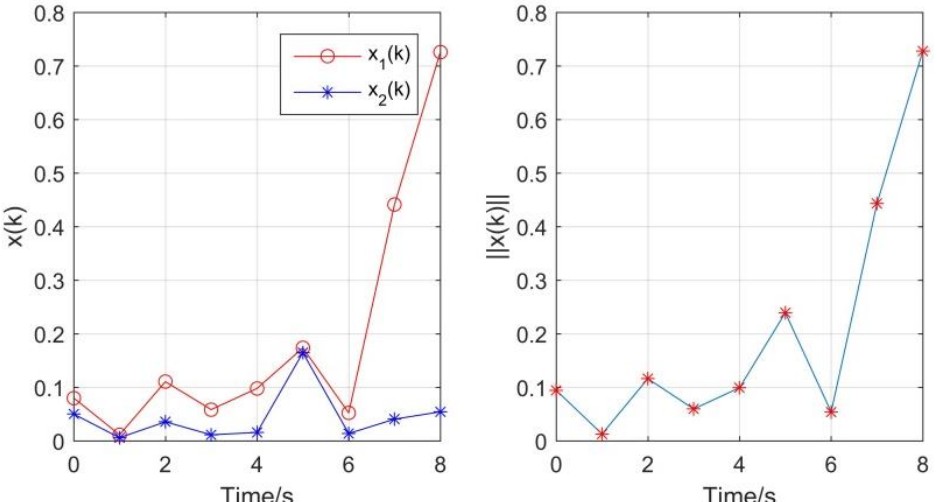

**Figure 6.** State trajectories of $x(k)$ (**left**) and trajectories of $\|x(k)\|$ (**right**).

Then the related parameters are given as $c_1 = 0.095$, $c_2 = 3.5$, $T = 8$, and we can directly obtain that $r = 0.045$. Suppose the switching number of $\sigma(k)$ over the set $\mathbb{N}_0^8$ is 6, i.e., $m = 6$. We can prove that system (19) satisfies the conditions in Theorem 1, by checking:

$$2(m+1)r = 0.63 < 1, c_2 > c_1 F_{\alpha,1}(2(m+1)r, T - k_0) = 0.095 \times F_{0.35,1}(0.63, 8) = 3.19845 < 3.5.$$

The corresponding system state $x(k)$ under the switching signal in Figure 7 is shown in Figure 8 (left), and it can be seen from the simulation Figure 8 (right) that the value of $\|x(k)\|$ will not exceed the given value $c_2 = 3.5$ within 0 to 8 s. Therefore, system (19) is finite-time stable with respect to the triplet $(0.095, 3.5, 8)$. Thus, the conducted simulations verify that the considered system is finite-time stable with respect to $(0.095, 3.5, 8)$ if it satisfies the given conditions in Theorem 1.

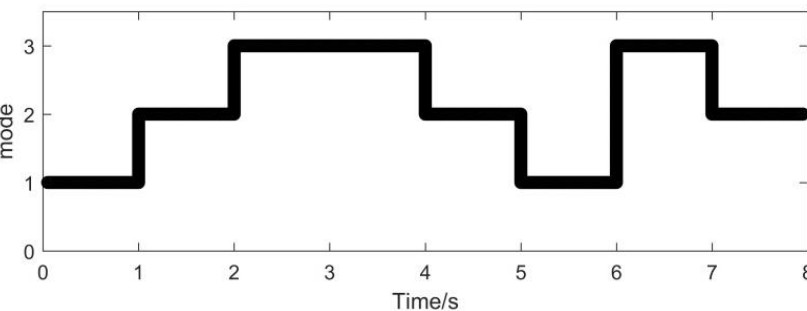

**Figure 7.** The switching signals $\sigma(k)$.

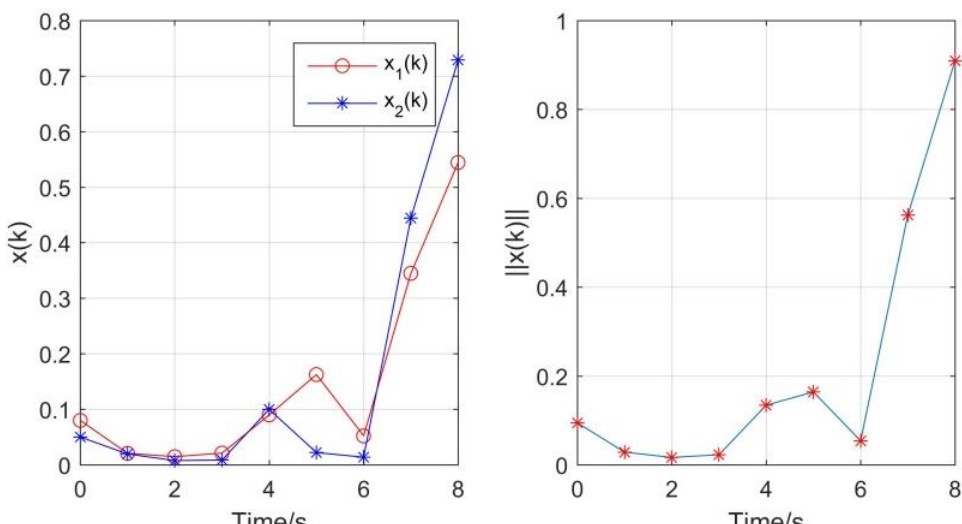

**Figure 8.** State trajectories of $x(k)$ (**left**) and trajectories of $\|x(k)\|$ (**right**).

Example 4 also shows that for the same system, when the conditions of Theorem 1 are satisfied, even if the switching signals are different, the system is still finite-time stable with respect to the triplet $(0.095, 3.5, 8)$.

## 5. Conclusions

We have investigated the finite-time stability problem of a class of Caputo nabla fractional-order switched linear systems with $\alpha \in (0, 1)$. Based on the novel solution expression for a Caputo nabla fractional-order switched linear system, the discrete Grönwall inequality and monotonicity of discrete Mittag-Leffler function, we have provided some sufficient conditions of finite-time stability for Caputo nabla fractional-order switched linear systems under the per-designed switching strategy. Furthermore, three numerical examples are used to illustrate the validity of the obtained results. Future work will try to investigate the stability of fractional-order discrete-time switched linear and nonlinear systems by using the multi-Lyapunov method.

**Author Contributions:** Conceptualization, P.X.; Data curation, P.X.; Formal analysis, P.X.; Funding acquisition, F.L.; Investigation, P.X.; Methodology, P.X.; Project administration, F.L.; Resources, P.X.; Validation, P.X.; Writing–original draft, P.X.; Writing–review and editing, P.X., F.L., Q.W., J.T., X.Y. and L.M. All authors have read and agreed to the published version of the manuscript.

**Funding:** This research was funded by [National Natural Science Foundation of China] grant number [61813006, 61973329], and funded by [the Key Projects of Basic Research Program of Guizhou Province] grant number [20191416], and [the Innovation Team of Universities in Guizhou Province] grant number [2022033].

**Institutional Review Board Statement:** Not applicable.

**Informed Consent Statement:** Not applicable.

**Data Availability Statement:** Not applicable.

**Acknowledgments:** This research was supported by the National Natural Science Foundation of China (61813006, 61973329), the Key Projects of Basic Research Program of Guizhou Province (20191416), the Innovation Team of Universities in Guizhou Province (2022033).

**Conflicts of Interest:** The authors declare no conflict of interest.

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
