# Peer review of "Finite-Time Stability for Caputo Nabla Fractional-Order Switched Linear Systems"

_fractalfract, doi:10.3390/fractalfract6110621_

Round 1

Reviewer 1 Report

Review Report on

Finite-time stability for Caputo nabla fractional order switched linear systems

Authors: Peng Xu  et al

The topic in this paper is new and very interesting, I give the following comments.

1)       The work contains some  typographical mistakes and errors and I suggest authors to read a complete paper and remove all typographical and grammatical errors.

2)       The authors are requested to add more details regarding their original contributions in this present work.

3)       The authors claimed proposed model, it should be changed.

4)       Some information is missing papers in the references and hence authors should need to modify it.

5) Authors need to write the advantage of the dynamical systems and its applications. In this connection, I directed the following papers; the authors cite the following articles in the introduction section:

a) Fractal and Fractional, 2022, 6(2), 61

c) Chaos, Solitons and Fractals, 2021, 146, 110915

5)       After most of the equations, mathematical punctuations are missing; authors should modify such a way that the work is free from these.  

6)       The introduction section needs to improve in a professional way.

7)       More description regarding the considered model needs to be present to improve the quality of the work.

Briefly, the Authors have derived some interesting results about the governing model. Therefore, it can be considered for publication after Minor Revision.

Reviewer 2 Report

Please see the attached reviewer report.

Reviewer 3 Report

The subject of the paper and the calculations are interesting, but the text has many drawbacks that make the article impossible to publish in its current form. In my opinion, the manuscript should be rethought and carefully edited and then sent as a new one to the editorial office.

Below, I present some noticed errors, inaccuracies, unclear wording, etc.:

The correct name of the basic Lemma used in the manuscript is "Gronwall inequality" and not "Gronwald inequality”,

The original version of the generalized Gronwall inequality (Theorem 3.1, pp. 860-861 C. Chen et al [26]) has more assumptions compared with the version cited by the Authors (Lemma 2.2[26], p.3). There should be given an explanation.

Similar calculations as on page 3 are also presented in the article [26], C. Chen et al, p. 860. It needs a comment.

In the section “Numerical examples” pp. 7-9 the Authors write: “This section uses the Matlab toolbox to verify the theoretical results proposed in the paper with three examples”. However, the Authors only provide the results without any explanations, calculations, Matlab codes, etc. Thus, the results cannot be verified by the reader.

The bibliography should be edited carefully. For example: what is “ence” in item 5), lack of pages in e.g., 6, 11), unclear 9), a mistake in 12). In almost every item one can find mistakes.

The paper contains a lot of linguistic errors that need to be corrected.

Round 2

Reviewer 2 Report

The paper has been properly prepared. It can be published as it is.

Reviewer 3 Report

Thank you for your hard work. I feel that now the paper can be published